# Fuzzy Comprehensive Evaluation of Human Work Efficiency in a High-Temperature Thermal-Radiation Environment

**Yingni Zhai [1,2,*], Xinta Wang [1], Haobo Niu [1], Xianglin Wang [1], Yangwen Nie [1,3] and Yanqiu Huang [2,4]**

1    School of Mechanical and Electrical Engineering, Xi'an University of Architecture and Technology, No.13 Yanta RD., Xi'an 710055, China
2    State Key Laboratory of Green Building in Western China, Xi'an University of Architecture and Technology, No.13 Yanta RD., Xi'an 710055, China
3    Basic Mechanical Experiment Center, Xi'an University of Architecture and Technology, No.13 Yanta RD., Xi'an 710055, China
4    School of Building Services Science and Engineering, Xi'an University of Architecture and Technology, No.13 Yanta RD., Xi'an 710055, China
*    Correspondence: ynzhai2013@xauat.edu.cn

**Abstract:** Work efficiency is not only related to objective cognitive performance, but it is also closely related to subjective psychological feelings. The existing research mostly focuses on the human safety tolerance limit from the physiological perspective, and there are few studies on work efficiency. In order to evaluate the work efficiency of humans in a high-temperature thermal-radiation environment, this paper proposes a method of personnel work efficiencies evaluation based on the fuzzy comprehensive evaluation method. In this method, the objective cognitive ability and subjective questionnaire data of human subjects in a thermal-radiation environment were obtained by building a high-temperature thermal-radiation platform, and the ergonomic status score (ESS) was calculated to determine the work efficiency evaluation grade. The results of this study indicate that subjective job satisfaction has a strong influence on work efficiencies. The safe WBGT temperature threshold that can be borne by personnel in a high-temperature thermal-radiation environment is lower than that in a single high-temperature environment. The ergonomic status score decreased significantly with the increase of radiation temperature. When the radiation temperature is 300 °C (WBGT temperature ≥ 33.2 °C), the work should be stopped immediately to ensure the safety and health of personnel. The research results can provide a reference for labor protection in high-temperature working environments.

**Keywords:** high-temperature thermal radiation; fuzzy comprehensive evaluation; work efficiency; wet-bulb black-bulb temperature (WBGT)

## 1. Introduction

High-temperature thermal radiation exists widely in metallurgy, ceramics, casting, mechanical thermal processing, and in other industries [1–3], and is a common hazard factor. Because a high-temperature environment has a negative impact on human body function, it also has a certain impact on people's mental state [4]. In a high-temperature heat-radiation environment, the error rate of perception and thinking ability increases, and alertness decreases [5], which affects work efficiency and personnel safety. Therefore, it is particularly important to study the work efficiency of humans in a high-temperature environment.

A person's work efficiency is closely related to his or her physiological and psychological state and cognitive performance. In 1998, Moran et al. [6] first proposed the human physiological stress index PSI based on rectal temperature and heart rate, mainly to evaluate human physiological stress in high-temperature environments. Zhang Jinggang et al. [7] studied the physiological and psychological effects of underground workers in

high-temperature and humid environments and found that with the increase of temperature and humidity, people's attention, reaction ability, and cognitive ability all declined. The high temperature and humidity environment aggravates fatigue and increases the error rate. Liu Weiwei [8] conducted a cognitive ability test at 37 °C and found that working continuously for 45 min under high temperature may have a negative impact on cognitive ability. Tian X [9] studied the cognitive performance of residents living in a subtropical zone under extremely high indoor temperatures and reduced humidity and found that the accuracy of these cognitive tests decreased significantly when the indoor temperature increased from 26 °C to 39 °C widely at 70% relative humidity.

In existing high-temperature operating environments, the proportion of mental work is increasing year by year. Due to the complexity of mental work, there is no unified measurement and evaluation method for its work efficiency. Wen Yi and Albert P. C. Chan [10,11] simulated the working process of workers in a high-temperature environment using simulation software and found the best working and rest time according to working efficiency, so as to provide a reference for work in real industrial workspaces. It has been pointed out that Wet Bulb Globe Temperature (WBGT) is the optimal index to evaluate the working efficiency at a high temperature. Lan L [12] studied the influence of the acoustic, light, and thermal environment on the working efficiency of personnel, and established a neurobehavioral capacity model to evaluate the working efficiency of personnel in indoor environments, making contributions to guiding actual industrial engineering operations. In 2019, Zheng et al. [13] introduced the fuzzy comprehensive evaluation method for the first time to evaluate the physiological state of human beings in an indoor high-temperature environment, established a quantitative comprehensive evaluation model of physiological state, proposed a safety rating of physiological state, and verified the model with an example. Although these studies involve high-temperature and thermal environments, most of them focus on physiological state and cannot effectively evaluate the efficiency of complex work such as mental work.

The existing research on work efficiency evaluation mostly focuses on the office environment, while the high-temperature thermal-radiation environment is relatively rare. Witterseh et al. [14] studied the work efficiency of personnel in open office environments with air temperature of 22/26/30 °C and noise of 35/55 dB (A). Greater noise and higher temperature can increase fatigue symptoms and reduce work efficiency. Hansen et al. [15] investigated the reactivity to cognitive tasks using heart rate (HR) and heart rate variability (HRV). The results showed that people with high HRV index showed more correct reaction, fewer errors, and faster average reaction time. Ye Xiaojiang et al. [16] studied the impact of indoor environment on workers' work efficiency, using the average count number of workers as an index to measure the size of work efficiency. Tiller et al. [17] studied the subjective perception and typing and digital inspection of task performance in an office environment with an air temperature of 18–26 °C and a noise of 30/40/50 dB (A). It was found that thermal comfort was affected by noise level, while ratings of building or office noise were not affected by the ambient temperature, and the task performance did not change significantly under several environmental conditions. There are few studies on work efficiency evaluation in high temperature environment, and most of them are for office environment.

In order to evaluate the work efficiency of personnel in a high-temperature working environment, this paper builds an evaluation model based on the fuzzy comprehensive evaluation method on the basis of the literature [5], combined with a subjective survey questionnaire and objective cognitive ability test data. The ergonomic state score is calculated, the work efficiency grade is evaluated, and the trend of change of the work efficiency grade with WBGT is analyzed. The research results can provide a basis for the effective protection of personnel in high-temperature heat radiation workshops.

## 2. Methodologies

### 2.1. Experimental Facilities and Conditions

The test was conducted in the laboratory of Xi'an University of Architecture and Technology from 1–30 June 2021. The room was arranged to simulate a high-temperature thermal-radiation environment, and it had sufficient airtightness and strong heat-insulation performance. The measurement points and subjects were placed symmetrically. During the test, light and wind speed were constant, and there was no other heat source and no other noise. In order to meet the high-temperature requirements, a workshop environment was simulated by using a high-temperature radiant plate. The test layout is shown in Figure 1.

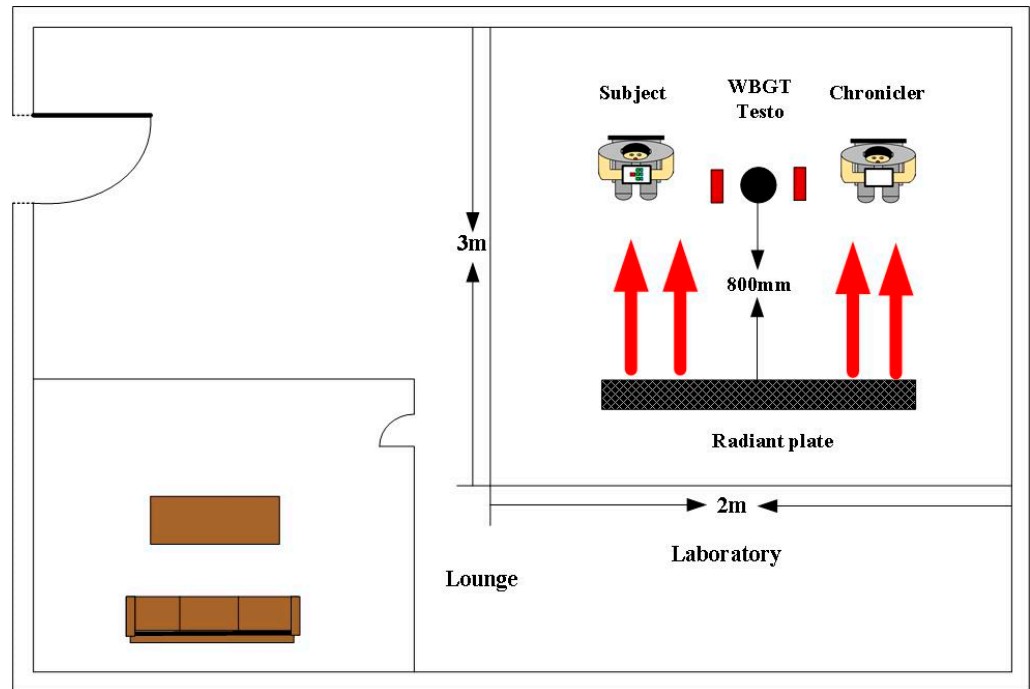

**Figure 1.** Layout of experimental room.

### 2.2. Participants

The within-subject design was adopted in this study, and the sample size was determined according to statistical power (Sp) and effect size (Es) [12]. In the domestic and foreign relevant literature concerning human trials, the number of subjects has mostly been between 8 and 16 [18–20]. In this paper, 12 undergraduates, aged 21–25 years, who volunteered to participate in the experiment, were selected as the subjects. It was ensured that each subject had adequate sleep, good living and eating habits, and had no history of disease. The test required uniform summer clothing (short-sleeved open-neck shirt and shorts); the thermal resistance of the clothing was about 0.4 clo (including the thermal resistance of the chair) [21,22]. All the subjects participated in five kinds of working conditions tests—a total of 60 groups. The Latin square balance test sequence and the influence of the fatigue effect were used, and the time was 13:00–14:45 and 15:30–17:15 each day.

### 2.3. Test Equipment

According to the ASHRAE standard 55-2017 [23], the test instrument was placed on the test bench at a height of 1.1 m, and the high-temperature radiation plate was 80 cm away from the subject. The environmental parameters in the test were measured by a WBGT index instrument, a high-temperature radiation plate, a temperature and humidity monitor, and other equipment, as shown in Table 1 below.

**Table 1.** Experimental equipment.

| Test Parameter Name | Equipment | Type | Range and Accuracy | Instrument |
|---|---|---|---|---|
| WBGT temperature | Index of WBGT-2006 m | WBGT-2206 | Accuracy of ±0.5 °C |  |
| Radiant panel | Curved surface high Temperature radiant electric heater | XLQ-4000 | Power 4000 W, rated frequency 50 HZ, 1500 mm × 390 mm |  |
| Environment temperature Relative humidity | Testo temperature and humidity meter | 605 i | −20–60 °C 0–100% |  |
| Wind speed | Testo anemometer | 405 i | Scope of 0–30 m/s |  |
| Body temperature | Non-contact infrared frontal temperature meter | DT-8806H | Distance: 1–15 cm, precision ±0.2 °C |  |
| Systolic and diastolic blood pressure and heart rate | Omron electronic blood pressure monitor | U12 | Scope of 0–39.9 kpa, precision ±0.4 kpa |  |
| Oxygen saturation, pulse | Pulse oximetry | Prince-100 A | Scope of 35–100% |  |

### 2.4. Experimental Protocol

Before the start of the test, the radiation plate was opened and stabilized to the target temperature, and the environmental parameters (indoor temperature, air velocity, WBGT, humidity) were measured after 30 min. The subjects sat quietly in the lounge for 20 min, and then physiological data measurement and a neurobehavioral ability test were carried out. After sitting quietly in the high-temperature thermal-radiation environment for 40 min, the neurobehavioral ability test and physiological data measurement were performed again. After the task was completed, a subjective questionnaire was filled in. The whole process of the test took 115 min in total. A flow chart is shown in Figure 2.

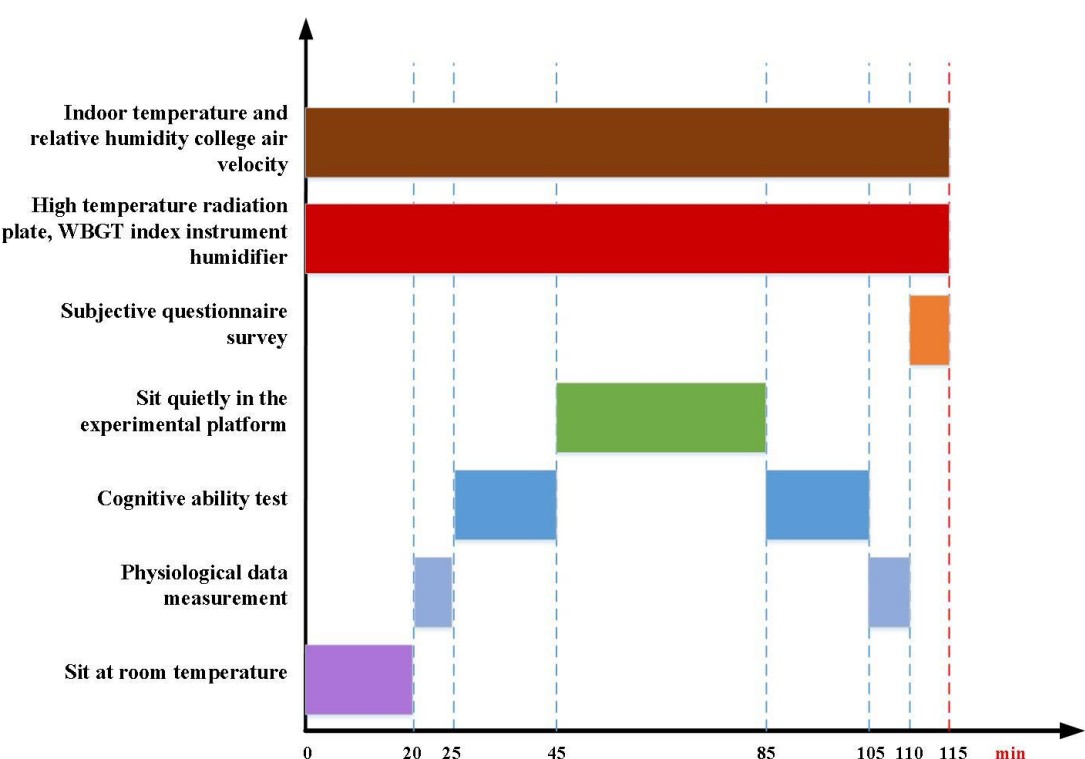

**Figure 2.** The procedure of experiment.

*2.5. Cognitive Test Tasks*

    In order to evaluate the influence of the thermal-radiation environment on humans, in this paper we selected several representative items for the neurobehavioral ability test [24], including perception, memory, thought, alertness, and others. Table 2 shows the detailed contents of the cognitive test.

**Table 2.** Cognitive test content.

| Testing Capability | Test Project | Test Content | Legend |
|---|---|---|---|
| Perception | Find the difference | Find the different graphics among 120 graphics Within the specified time | |
| | Find the target graphic | Find out the figure given by the topic from 36 different figures. | |
| Memory | Digital memories | A series of meaningless numbers are randomly given on the screen. Remember the numbers within the specified time and write them correctly. Write errors or timeouts are errors. | |
| | Matching memory | Pairwise pairing is done by flipping over the graph a limited number of times. An error will occur if the number of times is exceeded. | |

**Table 2.** *Cont.*

| Testing Capability | Test Project | Test Content | Legend |
|---|---|---|---|
| Thought | Rapid calculation | In a certain period of time, arrive at the answers of five mixed numbers through mental arithmetic and then select the correct option. |  |
| Alertness | 1 to N | Under the premise of allowing three chances for error, the subjects were asked to click from small to large among N discontinuous numbers. |  |

### 2.6. Subjective Survey Questionnaire

A survey questionnaire was conducted after the cognitive test. The content of the survey questionnaire comprised five parts: thermal sensation, environmental acceptance, work willingness, mental demand, and job satisfaction. Thermal sensation was designed based on the ASHRAE Standard 55-2017 [23]. Work willingness, mental demand, and job satisfaction of the subjects were evaluated by the NASA-TLX [25] scale.

### 2.7. Data Processing Method

In this study, the objective cognitive ability and subjective questionnaire data were statistically analyzed using *SPSS26.0.* For work efficiency evaluation, the cognitive ability data (total number of errors, total time) and subjective survey questionnaire data (work willingness, mental demand, job satisfaction) were selected as the evaluation indexes. A trapezoidal fuzzy distribution was adopted to construct the membership function of the evaluation indexes. The entropy weight method was used to determine the weight of the evaluation indexes, and the ergonomic state score (ESS) was introduced to construct the fuzzy comprehensive evaluation model.

## 3. Evaluation Model of Personnel Work Efficiency Based on the Fuzzy Comprehensive Evaluation Method

Human work efficiency is related not only to cognitive ability, but also to mental state. In this paper, the psychological-feeling data of the subjects were obtained through the survey questionnaire, which has certain subjectivity and uncertainty. Therefore, the membership function of fuzzy mathematics was used to convert the qualitative evaluation into a quantitative evaluation, and a fuzzy comprehensive evaluation model was established to evaluate work efficiency [26–29]. The specific steps were as follows.

### 3.1. Determine the Factor Set and Evaluation Set of the Work-Efficiency Evaluation

In the experiment, a neurobehavioral ability test was carried out, and the number of errors and the actual time spent by the subjects were recorded. The subjective feeling can be used as an evaluation index of work efficiency. Therefore, the factor set of work efficiency evaluation U = {objective measurement (U1), subjective evaluation (U2)}.

The factor set of objective measurement is:

U1 = {total number of errors (U11), total time (U12)}.

The factor set of subjective evaluation is:

U2 = {work willingness (U21), mental needs (U22), job satisfaction (U23)}.

The survey questionnaire shows that there is little difference in the thermal-sensation and environmental acceptance evaluation of the subjects, and so the factor set of subjective evaluation excludes the thermal-sensation and environmental acceptance evaluation.

The evaluation of personnel work efficiency was divided into four grades: high, medium, low, and very low. In order to quantitatively evaluate the work efficiency of personnel, a number was introduced as the evaluation scale corresponding to the work-efficiency grade [30]. The evaluation set V can be quantitatively described as: V = {4, 3, 2, 1}.

*3.2. Determination of Membership Function and Establishment of Fuzzy Matrix R*

The key to fuzzy comprehensive evaluation is to determine the membership degree of each factor in the evaluation factor set, that is, the membership function [31,32]. In this paper, we chose the "minimum ambiguity method" to determine the membership function. First, we calculated the value of the working-efficiency indexes of subjects under different temperatures (where working-efficiency evaluation indexes are the factor of factor set), as shown in Table 3. Then, according to the ergonomic evaluation grade classification and safety standards [33], it is divided into four grades (high, medium, low, very low). The standard value range of each index under different evaluation grades are shown in Table 4. Finally, we adopt a trapezoid (semi-trapezoid) membership function to calculate membership degrees of each factor [34]. Taking the total number of errors in the factor set as an example, the membership function of total error times under different evaluation grades are shown in Table 5, and other factors are similar.

**Table 3.** The values of work-efficiency indexes of subjects at different temperatures.

| Test Items | 100 °C | 150 °C | 200 °C | 250 °C | 300 °C |
|---|---|---|---|---|---|
| Total number of errors | 2.3 | 2.5 | 2.9 | 3 | 3.25 |
| Total time | 932.2 | 944.5 | 906.7 | 941.9 | 870.4 |
| Work willingness | 2.75 | 4.5 | 6.1 | 7.6 | 7.7 |
| Mental demand | 6.6 | 5.8 | 7.1 | 5.2 | 8.9 |
| Job satisfaction | 2.5 | 3.6 | 3.4 | 3.7 | 5.5 |

**Table 4.** The standard value range of indexes under different evaluation grades.

| Test Items | High | Medium | Low | Very Low |
|---|---|---|---|---|
| Total number of errors | [0–1) | [1–3) | [3–5) | [5–10) |
| Total time | [750–840) | [840–950) | [950–1100) | [1100–1300) |
| Work willingness | [0–2) | [2–5) | [5–8) | [8–10) |
| Mental demand | [0–2) | [2–5) | [5–8) | [8–10) |
| Job satisfaction | [0–2) | [2–5) | [5–8) | [8–10) |

**Table 5.** Membership function of total error times under different evaluation grades.

| Evaluation Grade | Membership Function |
|---|---|
| High | $y = \begin{cases} 1, x < 1 \\ \frac{3-x}{2}, 1 \leq x < 3 \\ 0, x \geq 3 \end{cases}$ |
| Medium | $y = \begin{cases} x - 0, x < 1 \\ 1, 1 \leq x < 3 \\ \frac{5-x}{2}, 3 \leq x < 5 \\ 0, x \geq 5 \end{cases}$ |
| Low | $y = \begin{cases} \frac{x-1}{2}, 1 \leq x < 3 \\ 1, 3 \leq x < 5 \\ \frac{7-x}{2}, 5 \leq x < 7 \\ 0, x \geq 7 \end{cases}$ |
| Very Low | $y = \begin{cases} 0, x < 3 \\ \frac{x-3}{2}, 3 \leq x < 5 \\ 1, x \geq 5 \end{cases}$ |

According to the constructed membership function and the standard value range of each index under different evaluation grades in Table 4, membership degrees under

different evaluation grades were calculated, and a single-factor evaluation matrix under different temperatures was obtained. Taking Subject 1 as an example (the calculation method of other subjects is the same as that of Subject 1), the single-factor evaluation matrix R (R1-1-1, R1-1-2, R1-1-3, R1-1-4, and R1-1-5) under different temperatures (100 °C, 150 °C, 200 °C, 250 °C, and 300 °C) is as follows:

$$
R_{1\text{-}1\text{-}1} = \begin{pmatrix} 0.5 & 1 & 0.5 & 0 \\ 0 & 0.62 & 1 & 0.38 \\ 0 & 1 & 1 & 0 \\ 0 & 0.67 & 1 & 0.33 \\ 1 & 0.5 & 0.2 & 0 \end{pmatrix} \quad R_{1\text{-}1\text{-}2} = \begin{pmatrix} 0 & 0.5 & 1 & 0.5 \\ 0 & 0.61 & 1 & 0.39 \\ 0.33 & 1 & 0.67 & 0 \\ 0 & 0.67 & 1 & 0.33 \\ 0 & 1 & 1 & 0 \end{pmatrix}
$$

$$
R_{1\text{-}1\text{-}3} = \begin{pmatrix} 1 & 1 & 0 & 0 \\ 0.85 & 1 & 0.15 & 0 \\ 0 & 0.33 & 1 & 0.67 \\ 0 & 0.67 & 1 & 0.33 \\ 0 & 0.67 & 1 & 0.33 \end{pmatrix} \quad R_{1\text{-}1\text{-}4} = \begin{pmatrix} 0.5 & 1 & 0.5 & 0 \\ 0 & 0.54 & 1 & 0.46 \\ 0 & 0.33 & 1 & 0.67 \\ 0 & 1 & 1 & 0 \\ 0 & 1 & 1 & 0 \end{pmatrix}
$$

$$
R_{1\text{-}1\text{-}5} = \begin{pmatrix} 1 & 0 & 0 & 0 \\ 0.71 & 1 & 0.29 & 0 \\ 0 & 0.67 & 1 & 0.33 \\ 0 & 0.33 & 1 & 0.67 \\ 0.67 & 1 & 0.33 & 0 \end{pmatrix}
$$

### *3.3. Weight Calculation of Work-Evaluation Factor Set*

Weight refers to the proportion of each factor in the factor set. This paper uses the entropy weight method to determine the weight of the evaluation index. The principle of the entropy weight method is to reflect the information contained in the index by the variation degree of the index. The smaller the degree of index variation, the less information is reflected and the lower is the corresponding weight [35]. The calculation steps of the entropy weight method are as follows:

① First, standardize the indexes. A positive index and a negative index have different meanings (a higher positive index value is better, and a higher negative index value is worse). Therefore, the positive index and negative index are standardized by different algorithms:

$$
\text{positive index} : \ x'_{ij} = \frac{x_{ij} - \min\{x_{1j}, \ldots, x_{nj}\}}{\max\{x_{1j}, \ldots, x_{nj}\} - \min\{x_{1j}, \ldots, x_{nj}\}}, \tag{1}
$$

$$
\text{negative index} : \ x'_{ij} = \frac{\max\{x_{1j}, \ldots, x_{nj}\} - x_{ij}}{\max\{x_{1j}, \ldots, x_{nj}\} - \min\{x_{1j}, \ldots, x_{nj}\}} \tag{2}
$$

For the calculation convenience, the normalized data is still recorded as $x_{ij}$.

② Calculate the proportion of the *i*-th sample value under the *j*-th index to the index:

$$
p_{ij} = \frac{x_{ij}}{\sum_{i=1}^{n} x_{ij}}, i = 1, \ldots, n, j = 1, \ldots, m. \tag{3}
$$

③ Calculate the entropy value of the *j*-th index:

$$
e_j = -k \sum_{i=1}^{n} p_{ij} \ln\left(p_{ij}\right), j = 1, \ldots, m, \tag{4}
$$

where k is a constant, $k = \frac{1}{\ln(n)} > 0$.

④ Calculate the difference of information entropy:

$$
d_j = 1 - e_j, j = 1, \ldots, m. \tag{5}
$$

⑤   Calculate the weight of each indicator:

$$w_j = \frac{d_j}{\sum_{j=1}^{m} d_j}, j = 1, \ldots, m. \tag{6}$$

Taking Subject 1 as an example (the calculation method for the other subjects is the same as that for Subject 1), its weight vector is as follows: A = (0.24, 0.18, 0.22, 0.13, 0.18).

*3.4. Comprehensive Evaluation Results*

According to the evaluation matrix and the weights of each evaluation index, the comprehensive evaluation results under different radiation temperatures (100 °C, 150 °C, 200 °C, 250 °C, and 300 °C) are calculated—taking Subject 1 as an example (the calculation method for the other subjects is the same as that for Subject 1).

$$B = A \circ R$$

where B is a comprehensive evaluation result vector; A is a weight vector determined according to the entropy weight method; R is a fuzzy matrix determined according to the membership function; "○" is a matrix composition operator.

We bring A and R1-1-1 into the formula to calculate the comprehensive evaluation result vector B1-1-1:

$$B_{1-1-1} = A \circ R_{1-1-1} = (0.3, 0.78, 0.74, 0.13).$$

Similarly, the comprehensive evaluation result vectors at other temperatures are as follows:

$$B_{1-1-2} = (0.07, 0.75, 0.93, 0.25)$$

$$B_{1-1-3} = (0.39, 0.73, 0.61, 0.27)$$

$$B_{1-1-4} = (0.12, 0.77, 0.88, 0.23)$$

$$B_{1-1-5} = (0.52, 0.69, 0.48, 0.14)$$

After normalization, the comprehensive evaluation result vectors under different temperatures are obtained as shown in Table 6.

**Table 6.** Vector of comprehensive evaluation results under different temperatures.

| Comprehensive Evaluation Result Vector | High | Medium | Low | Very Low |
|:---:|:---:|:---:|:---:|:---:|
| B1 | 0.12 | 0.45 | 0.43 | 0 |
| B2 | 0 | 0.4 | 0.5 | 0.11 |
| B3 | 0.13 | 0.5 | 0.37 | 0 |
| B4 | 0 | 0.43 | 0.5 | 0.07 |
| B5 | 0.3 | 0.43 | 0.27 | 0 |

The efficiency state score (ESS) is introduced to evaluate the safety grade of staff working efficiency.

$$ESS = \frac{\sum_{i=1}^{m} v_i b_i}{\sum_{i=1}^{m} b_i}, \tag{7}$$

where $v_i$ is the *i*-th element in evaluation set V and $b_i$ is the *i*-th element in comprehensive evaluation vector B.

According to the calculation formula of the ergonomic state score, the work-efficiency status score of subjects under different temperatures can be calculated, as shown in Figure 3. We can see from the figure that as the radiation temperature rises from 100 °C to 250 °C, the average score of the work-efficiency status decreases from 2.9 to 2.2. At 300 °C, due to individual differences in heat tolerance and heat adaptation [36], four subjects suffered from heatstroke, and only the remaining eight subjects with strong heat tolerance were tested. Therefore, the work-efficiency status score increased and is not used as a reference.

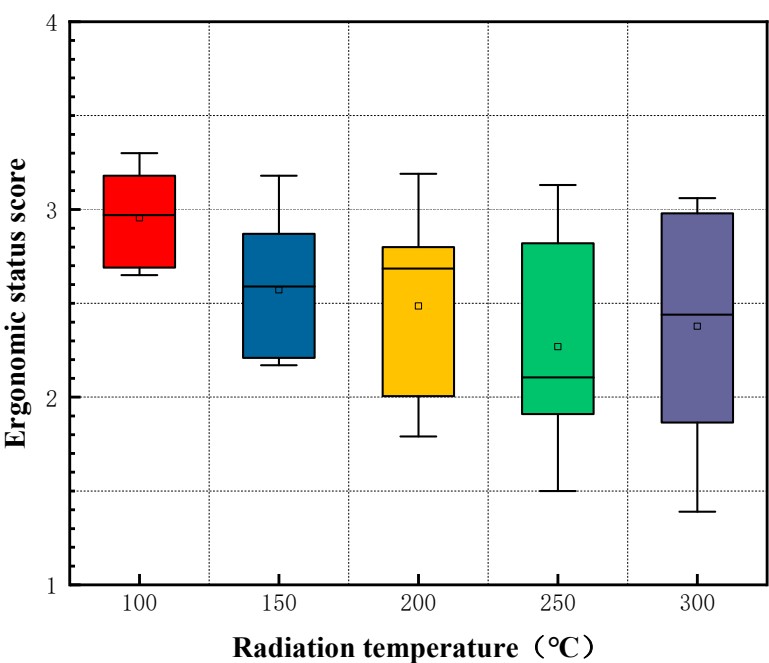

**Figure 3.** The ergonomic state score at each temperature level (4 people failed to complete the experiment at 300 °C).

According to the evaluation set V = {4, 3, 2, 1}, the scoring criteria of the work efficiency state grade in the high-temperature environment are shown in Table 7.

**Table 7.** Scoring criteria for ergonomic status.

| Index | Evaluation Grade | | | |
|---|---|---|---|---|
| | **High** | **Medium** | **Low** | **Very Low** |
| ESS | [3.0,4.0] | [2.0,3.0] | [1.0,2.0] | [0,1.0] |

Figure 4 shows the proportional distribution of work-efficiency evaluation grades under different radiation temperatures. We can see from the figure that when the radiation temperature increases from 100 °C to 150 °C, the work efficiency of personnel is in a medium or high state, in which high work efficiency decreases from 41.7% to 8.3%, and medium work efficiency increases from 58.3% to 91.7%. At 200 °C, 25% of the workers began to have a low working efficiency. The low working efficiency increased to 33.3% at 250 °C. When the radiation temperature was 300 °C, part of the personnel could not accept the high-temperature thermal-radiation environment and quit midway—because each individual's own thermal tolerance is different, and the radiation temperature exceeded the limit that the personnel could bear. Therefore, the work efficiency of different individuals shows obvious differences under a high-temperature heat-radiation environment.

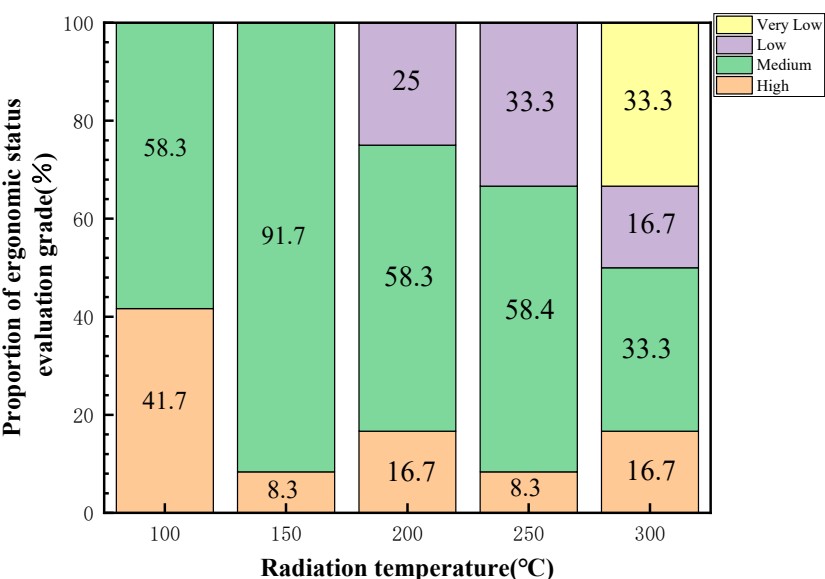

**Figure 4.** Evaluation grade of work efficiency under different temperature levels.

## 4. Discussion

In this paper, subjective feeling and objective cognitive ability are integrated to evaluate work efficiency, including work willingness, mental demands, job satisfaction, total number of errors, and total time. The entropy weight method is used to determine the weight of the work-efficiency index. The results are shown in Figure 5. We can see from the figure that the weight of job satisfaction and the total number of errors are relatively large. In existing work efficiencies research [37–39], only objective performance results are examined, and the subjective feelings of people are ignored. This study found that the subjective job satisfaction of personnel has a strong influence on work efficiencies. Therefore, in a high-temperature heat-radiation operating environment, it is of great significance to be attentive to the subjective psychological state of personnel and to improve job satisfaction for improving work efficiency.

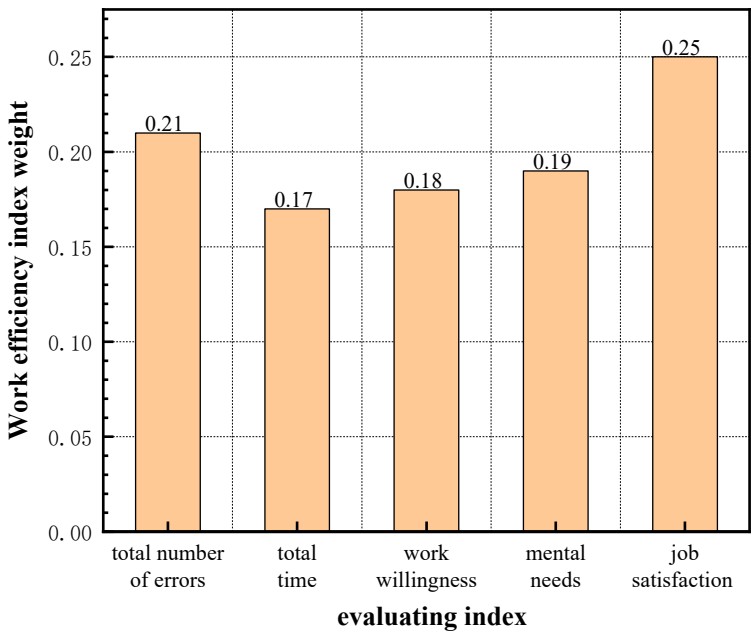

**Figure 5.** Work efficiency index weight under different temperatures.

Table 8 shows the WBGT temperature, heat radiation flux, indoor ambient temperature, and relative humidity under different radiation temperatures. It can be seen from the table that with an increase of radiation temperature, the temperature of WBGT and heat-radiation flux have an upward trend. When the radiation temperature is 100 °C, the temperature of WBGT has exceeded 25 °C, reaching the Class I high-temperature operation standard (WBGT ≥ 25 °C, indoor temperature ≥ 32 °C) [40].

**Table 8.** Indoor parameters under different working conditions.

| Radiation Temperature (°C) | 100 | 150 | 200 | 250 | 300 |
|---|---|---|---|---|---|
| WBGT (°C) | 28.21 ± 0.3 | 28.83 ± 0.5 | 29.94 ± 0.7 | 31.23 ± 0.6 | 33.16 ± 0.8 |
| Indoor temperature (°C) | 33.75 ± 0.4 | 34.8 ± 0.6 | 37.9 ± 1 | 40.8 ± 1.5 | 45.39 ± 1.4 |
| Relative humidity (%) | 46.58 ± 1.8 | 47.55 ± 1.7 | 44.34 ± 2.5 | 44.83 ± 3 | 42 ± 1.5 |
| Heat radiant flux (KW/m$^2$) | 0.19 | 0.32 | 0.5 | 0.75 | 1.1 |

"Workplace Occupational Disease Hazard Classification Part 3: High Temperature" GBZ/T229.3-2010 [41] stipulates that when the WBGT temperature is 36–38 °C, the exposure time to high-temperature work should be within 120 min for light-labor work, and rest should be arranged reasonably to reduce the exposure time of workers to high temperatures. In this study, we can see from Figure 4 that when the radiation temperature is 200 °C and the WBGT temperature is 29.9 °C, the work efficiency of the subjects begins to decrease significantly, and some subjects appear to be in a low work-efficiency state. At this time, the room air temperature is 37.9 °C. This is similar to the results in the literature [8], which point out that a high temperature at 37 °C damages the cognitive ability of subjects engaged in moderate-intensity activities. Therefore, in a high-temperature heat-radiation environment, when the WBGT temperature reaches 29.9 °C, it is recommended that labor-protection measures are taken. This temperature is lower than the "Workplace Occupational Disease Hazard Operation Classification Part 3: High Temperature" GBZ/T229.3-2010 Class II Operation Grade Labor protection temperature.

"Workplace Occupational Disease Hazard Classification Part 3: High Temperature" GBZ/T229.3-2010 [41] stipulates that in light-labor work of Class III operation grade, the exposure time to high temperature work should be within 120 min, and the WBGT temperature is 38–40 °C. Therefore, thermal-stress monitoring of workers should be carried out, and protective measures should be taken. When the temperature of the WBGT exceeds 40 °C, this is classified as Class IV, and the operation should be stopped immediately.

It can be seen from Figure 4 that when the radiation temperature is 300 °C, there were four subjects who did not complete the test, the work-efficiency status score was 0, and the work-efficiency status evaluation grade was "extremely low." Figure 6 shows the work-efficiency status scores of four subjects under different ambient temperatures. We can see from the figure that when the radiation temperature is 300 °C, and the WBGT temperature is close to 33.2 °C, the work efficiency of the four subjects decreased sharply and they felt obvious discomfort, resulting in heatstroke. In order to protect the safety and health of the personnel, the test was terminated immediately. In the literature [42], it is pointed out that when the temperature of WBGT is 38 °C, the operation should be stopped immediately, and heavy workload activities are not allowed [42]. When the WBGT is 33.2 °C, the possible reasons of the subjects suffering from heatstroke are that the subjects were exposed to both high temperature and strong thermal radiation during the test. The combined effect of these two factors caused the subjects to experience a higher heat load than they would in a single high-temperature environment. Therefore, it is recommended that the operation should be stopped immediately when the WBGT temperature exceeds 33.2 °C and Operation Class IV is reached in a high-temperature thermal-radiation environment.

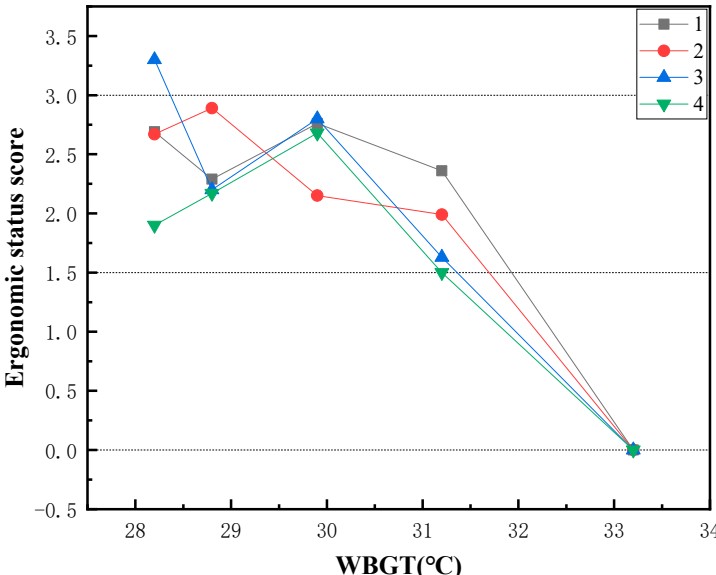

**Figure 6.** Ergonomic-status scores under different ambient temperatures (4 persons who failed to complete the test at 33.2 °C).

## 5. Conclusions

In order to study the impact of a high-temperature thermal-radiation environment on the work efficiency of personnel, a fuzzy comprehensive evaluation model was constructed by combining subjective questionnaires and objective cognitive data. The ergonomics status score (ESS) was calculated, the trend of variation of the ESS with WBGT was analyzed, and the influence of high-temperature thermal radiation on work efficiency was explored. The results are as follows:

(1) The order of the weight of the evaluation indicators affecting work efficiency is: job satisfaction, total number of errors, mental demands, work willingness, and total time. The subjective job satisfaction of personnel has a strong influence on work efficiency. Thus, for improving work efficiency, it is of strong significance to pay attention to the subjective psychological state of personnel and to improve job satisfaction.

(2) In a high-temperature thermal-radiation industrial environment, there is a negative correlation between work efficiency and WBGT temperature.

(3) Compared with "Workplace Occupational Disease Hazard Classification Part 3: High Temperature" GBZ/T229.3-2010, in a high-temperature heat-radiation environment, the threshold of a safe WBGT temperature that can be borne by personnel in a high-temperature thermal-radiation environment is much lower than the threshold of a single high-temperature environment. When the temperature of WBGT exceeds 29.9 °C, a state of low work efficiency occurs, and certain labor-protection measures should be taken at this time. When the WBGT is close to 33.2 °C, some subjects get heatstroke, and the operations in progress should be stopped immediately to ensure the safety and health of personnel.

The test subjects in this paper were undergraduates, and there are certain differences in an actual thermal processing workshop for personnel in terms of heat tolerance, heat acclimation, and work properties. In future research, we aim to add more trials and different types of personnel to evaluate work efficiency in actual workshop. Longer exposure experiments and more participants will also be carried out in our future research. At the same time, the research group also plans to do experimental research on other heat source distances in the future to analyze the impact of different heat source distances on human work efficiency. In addition, in the future experiment, we will expand the age range and education level of the subjects, and conduct a deeper study on the impact of thermal radiation on people's work efficiency.

**Author Contributions:** Conceptualization, Y.Z. and X.W. (Xinta Wang); methodology, Y.Z.; data curation, X.W. (Xinta Wang) and H.N.; writing—original draft preparation, X.W. (Xinta Wang); writing—review and editing, X.W. (Xinta Wang) and X.W. (Xianglin Wang) and Y.N. and Y.H. All authors checked the results and finalized the manuscript. All authors have read and agreed to the published version of the manuscript.

**Funding:** This study was supported by the National Key Research and Development Program of China (grant numbers No.2020YFE0200300).

**Institutional Review Board Statement:** The study was conducted in accordance with the Declaration of Helsinki. All subjects gave their informed consent for inclusion before they participated in the study.

**Informed Consent Statement:** Informed consent was obtained from all subjects involved in the study. Written informed consent was obtained from the patient(s) to publish this paper.

**Data Availability Statement:** The data presented in this study are available on request from the corresponding author.

**Conflicts of Interest:** The authors declare no conflict of interest.

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
