# Peer review of "Fuzzy Comprehensive Evaluation of Human Work Efficiency in a High-Temperature Thermal-Radiation Environment"

_sustainability, doi:10.3390/su142113959_

Round 1

Reviewer 1 Report

The subject sounds interesting. However the paper needs  revisions.

 Please read my comments/suggestions given below for preparing the revised draft:

1-The abstract is not convincing and is disorganized, it should be refined to precisely illustrate what authors have done in this paper. The abstract must be a concise yet comprehensive reflection of what is in your paper. Remember that reader want to know:  a-what is the problem. b- why the problem is relevant c- wants an overview of your approach. d-need to know the results.

2-Manuscript needs a good introduction, the introduction section of the manuscript is weak, authors are advised to improvise the introduction section.

3- Section 3.4 is weak. Give numerical comparison with existing methods.

4. Expand literature by citing related work: 

A fuzzy climate decision support systems for tomatoes in high tunnels, International Journal of Fuzzy Systems, 19(3)(2017):751-775.

Fuzzy decision support system for fertilizer, Neural Computing & Applications, 25(6)(2014) 1495-1505.

A novel fuzzy decision making system for CPU scheduling  algorithm, Neural Computing & Applications, 27(7)(2016) 1927-1939.

Intuitionistic Fuzzy Logic Control for Heater Fans, Mathematics in Computer Science, 7(3)(2013) 367-378.

Author Response

请参阅附件。

Reviewer 2 Report

The authors propose a method of personnel work efficiency evaluation based on the fuzzy comprehensive evaluation method to evaluate the work efficiency of humans in a high-temperature thermal-radiation environment. The results of this study indicate that subjective job satisfaction has a strong influence on work efficiency. The safe WBGT temperature threshold that personnel can bear in a high-temperature thermal-radiation environment is lower than that in a single high-temperature environment. The research results provide an interesting reference for labor protection in high-temperature working environments. The paper is a well-written scientific work. However, before accepting, some shortcomings must be eliminated. The list of my comments is as follows:

1. Both the abstract and introduction sections should emphasize the contribution. Also, more emphasis should be given to the motivation and justification for undertaking research in this direction.

2. Captions and figures should be on the same page (always).

3. Table 5. How were these membership functions identified?

4. Lines 189-191. Please fix the lower indexes. There should be more clear and please avoid "_". Itis also used in the rest parts of the paper. Please fix it. 

5. Line 206. Instead, "Ccalulate" should be "Caluclate". Please check the whole manuscript. English should be improved - native speaker assistance should be considered.

6. Line 222. What is this operator? Is this an algebraic product?

7. In figures, please add a grid (grid on).

8. Please add future research directions.

9. The references need to be updated and extended. The references should be in one style; please unify them.

Author Response

请参阅附件。

Round 2

Reviewer 1 Report

I accept revised version. 

Author Response

Thank you!

Reviewer 2 Report

The authors only partially improved the paper.  Future research directions must be extended. Moreover, Fig 1 is too low quality and unprofessional. The most important is the issue of fuzzy set identification. The authors say:

"We adopt a trapezoid (semi-trapezoid) membership function to calculate membership degrees of each factor. This is shown in Section 3.2 in red color. The key to fuzzy evaluation is to calculate the degree of membership of the secondary index U i j to each grade V i in the evaluation set. That is, the membership function is established. Taking the total number of errors in the factor set as an exampl"

The question was how it was identified. I do not see the answer which Is crucial for this research. It was should be describe step by step in the paper.

Round 3

Reviewer 2 Report

The paper has been improved. Figures should be improved because the quality is very poor.

Author Response

请参阅附件。
